**Data Availability Statement:** Data are currently publicly available in Gene Expression Omnibus (GSE237142). Data can be retrieved online via https://www.ncbi.nlm.nih.gov/geo/ by using the

# Multiplexed assay of variant effect reveals residues of functional importance in the *BRCA1* coiled-coil and serine cluster domains

Gregory Nagy[1], Mariame Diabate[1], Tapahsama Banerjee[1], Aleksandra I. Adamovich[1¤a], Nahum Smith[2], Hyeongseon Jeon[1], Shruti Dhar[1], Wenfang Liu[1¤b], Katherine Burgess[1], Dongjun Chung[1], Lea M. Starita[2], Jeffrey D. Parvin[1]*

**1** Department of Biomedical Informatics, The Ohio State University Comprehensive Cancer Center, Ohio State University, Columbus, Ohio, United States of America, **2** Department of Genome Sciences, University of Washington and Brotman Baty Institute for Precision Medicine, Seattle, Washington, United States of America

¤a Current address: Laboratory of Genitourinary Cancer Pathogenesis, Center for Cancer Research, National Cancer Institute, National Institutes of Health, Bethesda, Maryland, United States of America
¤b Current address: Department of Molecular Genetics, College of Arts and Sciences, Ohio State University, Columbus, Ohio, United States of America

* Jeffrey.Parvin@osumc.edu

## Abstract

Delineating functionally normal variants from functionally abnormal variants in tumor suppressor proteins is critical for cancer surveillance, prognosis, and treatment options. BRCA1 is a protein that has many variants of uncertain significance which are not yet classified as functionally normal or abnormal. *In vitro* functional assays can be used to identify the functional impact of a variant when the variant has not yet been categorized through clinical observation. Here we employ a homology-directed repair (HDR) reporter assay to evaluate over 300 missense and nonsense BRCA1 variants between amino acid residues 1280 and 1576, which encompasses the coiled-coil and serine cluster domains. Functionally abnormal variants tended to cluster in residues known to interact with PALB2, which is critical for homology-directed repair. Multiplexed results were confirmed by singleton assay and by ClinVar database variant interpretations. Comparison of multiplexed results to designated benign or likely benign or pathogenic or likely pathogenic variants in the ClinVar database yielded 100% specificity and 100% sensitivity of the multiplexed assay. Clinicians can reference the results of this functional assay for help in guiding cancer treatment and surveillance options. These results are the first to evaluate this domain of BRCA1 using a multiplexed approach and indicate the importance of this domain in the DNA repair process.

## Introduction

Determining the change in protein function due to missense variants of tumor suppressor proteins and oncoproteins is critical for cancer surveillance and treatment. Knowing which variants induce hypo- or hyperfunctional protein activity can guide treatment decisions; for example, knowing the functional impact of a variant in *BRCA1* can impact the level of pre-

Gene Expression Omnibus search function with the search value GSE237142.

**Funding:** JDP and LMS were supported by R01 CA228083 from the National Cancer Institute. MD was supported by R01 supplement CA228083-01A1S1 from the National Cancer Institute GN was supported by T32 GM068412 from the National Institute of General Medical Sciences and a Pelotonia Fellowship. The funders had no role in study design, data collection and analysis, decision to publish, or preparation of the manuscript.

**Competing interests:** The authors have declared that no competing interests exist.

cancer surveillance or prophylactic surgeries and may affect cancer care decisions such as including DNA-damaging agents or PARP inhibitors in the treatment regimen [1]. Sometimes, the functional impact of a protein variant is known, and confirmation of the protein variant through sequencing can guide treatment. Many times, however, the functional impact of a missense variant is unknown, and these are aptly described as variants of uncertain significance (VUS) [2, 3]. VUS pose complications for patients as these individuals cannot receive information regarding their risk of disease or their treatment options. Additionally, VUS occur more frequently in sequencing results of underserved populations [4, 5]. It is important to cancer care to identify the functional impact of a VUS and whether it is likely to predispose to disease.

When the functional impact of VUS cannot be deduced from family-based or population data, *in vitro* functional assays can provide the needed information. Testing one variant at a time is resource- and time-intensive. Such an approach is also backward-looking and functional results would not be available when the variant was identified. By contrast, multiplexed functional assays, in which hundreds of protein variants are tested simultaneously, can fill in the gap to determine the functional impact of protein variants, and the functional effects of sequence variants can be obtained before any are seen in the clinic.

*BRCA1*, for which germline carriers of pathogenic variants face significant lifetime hereditary breast and ovarian cancer risks [6], frequently presents as a VUS in individual sequencing results. Identification of a pathogenic *BRCA1* variant through sequencing can guide clinical decisions, including prophylactic measures such as mastectomy or oophorectomy [1], or can guide treatment with platinum, PARP inhibitors, or topoisomerase inhibitors [7–9]. As of May 2023 there are 1700 *BRCA1* VUS in the ClinVar database [10]. VUS frequently appear in the amino-terminal RING domain, the carboxy-terminal BRCT domain, and the coiled-coil domain near the middle of the protein. While multiplexed functional assays have been conducted on the RING and BRCT domains [11–15], to our knowledge there has yet to be a multiplexed functional assay on the coiled-coil domain and residues surrounding it. Here we present a multiplexed functional assay of BRCA1 missense and nonsense variants between residues 1280 and 1576, which encompasses the serine cluster domain and the coiled-coil domain, known to bind to PALB2 [16]. In this 297-residue region there are more than 350 missense VUS in ClinVar.

The biochemical function of BRCA1 that is perhaps most essential for its tumor suppression activity is the regulation of repair of DNA double strand breaks via the homology directed repair (HDR) mechanism [17, 18]. In the current study, we generated libraries of BRCA1 mutated at single amino acid residues from 1280–1576 and analyzed the function of these variants in the HDR assay. Overall, more than 300 variants were evaluated in the multiplex assay with high sensitivity and specificity.

While reconciling multiple sources of data, including population data, case-control studies, computational predictions, and well-established *in vitro* functional assays like the one used in the current study, the interpretation of the effects of BRCA1 variants on breast and ovarian cancer predisposition can be difficult. A framework was developed [19] to guide the interpretation of the significance of variants from the diverse data. Updates to this framework have been described by CanVIG-UK [20] and ClinGene/ENIGMA [21]. In this study, we apply a well-established *in vitro* assay to measure the functional impact of a variant in BRCA1, and this value is used to infer whether a variant is benign or pathogenic. In the standard framework, the evidence for pathogenicity from *in vitro* studies has the prefix PS3 and the rating very strong, strong, moderate, or supporting, depending on the strength of the correlation of the measured function of variants known to be pathogenic (loss of function). Similarly, the evidence for benignity has the prefix BS3 and ratings of strong, moderate, or supporting depending on the strength of correlation of measured function with known benign variants

(functionally normal). Previous applications of the assay for function of BRCA1 in DNA repair for variants in the amino-terminal domain [12] or the carboxy-terminal domain [13] yielded classifications of PS3-moderate and BS3-moderate to BS3-strong for the strength of pathogenicity or benignity, respectively.

## Methods

### Plasmid construction

The BRCA1 encoding residues (GenBank: AY888184.1) [22] 1243 to 1863 (CC1) or 1294 to 1863 (CC2 and CC3) was inserted into pUC19 for mutagenesis reactions, and the variant library encoding the BRCA1 fragment was then reinserted into a vector for the full length expression of BRCA1 proteins carrying a single missense substitution. Inverse PCR [23] was performed on the pUC19-BRCA1Cterm plasmid using mutagenic primer libraries (S5 Table in S1 Data) to make linear PCR product with a single amino acid substitution. Linear PCR products were pooled according to relative brightness on EtBr gel so that each codon has approximately equal representation in the library. The three mutagenic libraries total 297 codons: CC1 encoded variants in residues 1280–1375; CC2 encoded variants in residues 1376–1471, and CC3 encoded variants in 1472–1576. The pooled library was digested with Dpn1 and the product was subsequently gel purified. The gel-purified product was phosphorylated with T4 PNK and subsequently self-ligated with T4 DNA ligase and transformed into DH5alpha MAXefficiency cells (Invitrogen). A small amount (10 µL) of the transformed bacterial cells were plated onto an Ampicillin agar plate for counting the concentration of colonies, and the remainder of the cells were grown overnight in a 250 mL culture. The 250 mL culture was frozen in Buffer P1 (Qiagen) until confirmation of adequate percentage of missense substitutions through Sanger sequencing. Twenty single colonies were sampled from the library and were sequenced; over 70% of the plasmids were correct. After sequencing confirmation, the DNA of the plasmid library was purified to yield a pUC19-BRCA1-CCmut plasmid library. The plasmid libraries and details about subcloning are available from the authors.

From the pUC19-BRCA1CCmut libraries, the BRCA1 portion was excised by PCR using the primers, BRCA1-NheI-fwd-longer-GN and BRCA1-longer-Sbf1-Barcode-rev (S6 Table in S1 Data), for CC3 and CC2 and combined with pcDNA5-BRCA1-Blue cut with Sbf1 and Nhe1 using NEB HiFi DNA Assembly master mix to make the full length BRCA1 expression vector.

### Long-read sequencing to pair barcodes with variants

pcDNA5 variant library for CC2 and CC3 was excised by digestion with Sbf1 and NheI and the ~1800 bp fragment was gel purified, and the CC1 variant library was similarly excised but using enzymes Nsi1 and Sbf1, and gel purified. All three libraries were prepared using PacBio SMRTBell Prep Kit 1.0. CC1 yielded 792 unique single substitution variants between codons 1280 and 1576. CC2 yielded 1092 unique single substitution variants between codons 1280 and 1576. CC3 yielded 2041 unique variants. CC3 gave an expected yield, but CC1 and CC2 yielded a relatively low number of unique single substitutions. This led to relatively low number of variants integrated into HeLa-DR-FRT cells and subsequently low number of variants evaluated after read count cutoffs were applied.

### Integrating libraries into HeLa-DR-FRT cells

HeLa-DR-FRT cells, derived from HeLa-S3 cells (ATCC catalog CCL-2.2) and confirmed by short tandem repeat (STR) fingerprinting, were used for evaluation of plasmid libraries. For each 10 cm plate of HeLa-DR-FRT cells, 20 µg of pOG44 (Invitrogen) and 10 µg of the

pcDNA5 variant library was transfected. Integrated cells were selected using hygromycin (550 μg/mL). Colonies were counted in subsections of the plates, and a total colony count was deduced. Each cell contains and expresses a single variant at the FRT recombination site. For integration of the three variant libraries into the HeLa-DR-FRT cells, a minimum of 134,000 colonies per library were counted post-hygromycin selection.

## Homology-directed repair multiplexed assay

After seeding in 15 wells of a 24-well plate on day 0, cells were transfected on day 1 with 1.5 μL of Oligofectamine (Invitrogen) and 30 pmol of siRNA with one of three options: control siRNA (6 wells), siRNA to endogenous BRCA1 using an siRNA specific for the 3'UTR (8 wells), and siRNA to coding sequence (CDS) of BRCA1 (1 well). siBRCA1 CDS would inhibit translation of both the integrated and endogenous BRCA1. On Day 2, cells were transferred from a 24-well plate to a 6-well plate. On Day 3, cells were transfected with 3 μL Lipofectamine 2000 (Invitrogen) per well, along with 50 pmol of siRNA and 3 μg of plasmid encoding the I-Sce1 restriction enzyme. On Day 4, cells were transferred from 6-well plate to either 10 cm or 15 cm plate, with the exception of one well from each of the three treatments, which were kept in 6-well plate for fluorescent evaluation on Day 5. On day 5, one sample from each treatment group was run on the Calibur Flow Cytometer (S1 Fig, S7 Table in S1 Data). If the GFP-positive % for the siRNA to the endogenous BRCA1 was lower than the GFP-positive % of the control siRNA sample, the experiment proceeded. On day 6, cells were resuspended in sorting buffer (DPBS with 5 mM EDTA, 25 mM HEPES, 1% FBS previously dialyzed in DPBS). Cells were sorted based on GFP fluorescence, and genomic DNA was subsequently extracted to identify the integrated BRCA1 variant in each cell. After saving 10% of the cells as an input sample, the cells were sorted on the ARIA III in the flow cytometry core into pools of GFP-positive and GFP-negative cells. A minimum of $3.68*10^5$ cells were collected for each of the positive and negative cohorts (S9 Table in S1 Data). After sorting, gDNA was extracted from the cells using the DNEasy kit and eluted into 200 μL Buffer EB (Qiagen). The frequencies of variants in the GFP-positive and input cohorts were determined and a functional score for each variant relative to WT was calculated based on variant distribution between the GFP-positive and input cohorts using Enrich2 software [24].

Of 5940 possible variants (297 residues x 20 variants/residue), 3925 variants (66%) were created, 889 (23% of created variants) were integrated into the HeLa-DR-FRT cells, and of the 889 variants, 310 (35%) were evaluated after final read count cutoffs were established.

## Barcode amplification from gDNA and nested PCR

Extracted gDNA from the GFP-positive and GFP-negative bins of the multiplex assay was quantified by Qubit 4 Fluorometer, then barcodes were amplified using a nested PCR reaction. Either the entire gDNA sample or gDNA from $5.0*10^5$ cells was input into the reaction, whichever was less. The first PCR used primers Cterm56-Nest1-Fwd and FRT-LacZ-Rev for 22 cycles with Kapa 2G Robust (Roche). After the first reaction, products were bead purified with 0.6x AMPure PB beads (PacBio), quantified by Qubit, and 2 μL of product was analyzed on a TapeStation BioAnalyzer for size confirmation. After size confirmation, PCR products were subject to a second PCR reaction for 7 cycles to add Illumina index sequences. Indexed PCR products were purified first by 0.6x AMPure PB bead purification, in which the contaminants were captured by the beads and desired PCR product was left in the supernatant. The supernatant of the 0.6x bead purification was subject to 0.8x AMPure bead purification, and product was eluted into 10 mM Tris pH 8. Again, 2 μL of indexed PCR product was analyzed using a

TapeStation BioAnalyzer for size confirmation. Indexed samples were pooled once all Nest2 products were ready.

### Analysis of long-read and short-read sequencing

Variants were paired to barcodes using the PacBio code available on the Parvin Lab GitLab page: https://gitlab.com/jeffparvin/brca1_hdr_analysis.git. Barcodes paired with a cDNA sequence that had multiple amino acid changes were changed to WT except within the region of mutagenesis. For the assay, barcodes were counted and the score was compiled in Enrich2 [24], and duplicate variants were combined using scripts available on the Parvin Lab GitLab.

### Singleton HDR

HeLa-DR-FRT cells were plated on Day 0 into 24-well plate. On Day 1, 5 pmol of siRNA was transfected along with 0.3 μg of plasmid DNA. On day 2, cells were transferred from 24-well plate to 6-well plate. On day 3, 25 pmol siRNA was transfected concurrently with 0.75 μg of plasmid BRCA1 DNA and 0.75 μg plasmid for expression of I-Sce1. Cells were evaluated for GFP-positive % on Day 6 using Calibur or Fortessa flow cytometer.

### Western blot

Protein was extracted from cells using 0.1% NP-40, 50 mM Hepes, pH 7.4, 0.15 M NaCl, 5 mM EDTA, 10% glycerin. Protein samples were quantified through Bradford assay then subjected to electrophoresis on 5 or 6% fixed SDS-PAGE gel. Samples were then transferred onto PVDF membrane that had been activated with methanol before transfer. Primary antibodies used were BRCA1 (1:500) [25] and RHA1 (1:20,000) [25] in PBS with 1% BSA. Secondary antibody was diluted into 5% milk in blocking buffer, anti-rabbit (1:5000) HRP-conjugated secondary antibody (Cytiva) and visualized under chemiluminescent setting on LiCOR Odyssey Fc machine.

### qRT-PCR

Cells were resuspended in TRIzol and RNA was extracted using chloroform. Isolated RNA was treated with DNAse I and subsequently reverse transcribed. cDNA was run in qPCR using iQ SYBR Green Supermix (Bio-Rad) using primers to either the 3'UTR of endogenous, but not integrated, BRCA1, or to 18S rRNA as control (S8 Table in S1 Data).

## Results

### Multiplexed homology-directed repair assay

We performed a multiplexed assay to evaluate the function of BRCA1 missense and nonsense variants from residues 1280–1576 for homology-directed repair (HDR) of DNA double-strand breaks (Fig 1A). This region contains a serine cluster domain as well as the coiled-coil domain; the serine-cluster domain is phosphorylated following DNA damage [26–31] and the coiled-coil domain is known to bind to PALB2, an important step for the homology-directed repair pathway [16, 32–34]. We generated three pools of variants of about 100 codons each (referred to as CC1, CC2, and CC3) in a plasmid designed to express full-length BRCA1. This 297-codon region is immediately upstream of the region mutagenized and evaluated in a previous publication that measured the functional effects of missense and nonsense variants in the BRCT domain [13]. The three mutagenic libraries were each integrated into a derivative of the HeLa cell line called HeLa-DR-FRT. HeLa-DR-FRT cells have integrated in the genome a single cassette with inactive GFP direct repeats (DR) that are the substrate for measuring HDR activity [17], and these cells contain in the genome a single Flp-In Recombination Target

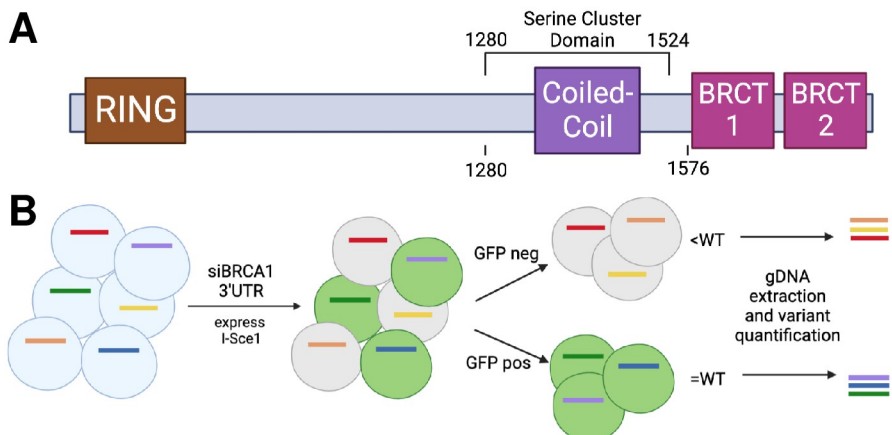

**Fig 1. BRCA1 protein domains and multiplexed assay schematic. A.** The BRCA1 protein contains an amino-terminal RING domain that binds with BARD1, a coiled-coil domain that binds with PALB2, and two carboxy-terminal BRCT domains that bind with a number of proteins that function in the DNA repair process. Deep mutational scans of the RING and BRCT domains have previously been published [11–15], and this study performs a deep mutational scan of the coiled-coil domain, as well as codons immediately N-terminal and C-terminal of the coiled-coil domain. Image created with BioRender.com. **B.** Schematic of multiplexed homology-directed repair assay. After library integration, each cell contains endogenous BRCA1 and a single integrated BRCA1 variant. Cells are treated with (1) a control siRNA or an siRNA that selectively silences the endogenous BRCA1 and (2) a plasmid encoding the I-SceI protein that initiates a DNA double-stranded break in a GFP reporter gene. In the control siRNA condition, the endogenous BRCA1 protein is expressed and will repair the double-stranded break. In the condition in which the siRNA selectively targets endogenous BRCA1, the integrated BRCA1 variant is left to repair the double-stranded break. Functionally normal variants will repair the double-stranded break at WT-like levels turning cells GFP-positive, and cells with nonfunctional variants will remain GFP-negative. Cells are then sorted into GFP-positive and negative cohorts. After sorting, gDNA is extracted from the cells and barcodes are extracted through PCR. Barcodes are sequenced, and the abundance of variants in the pool of cells is quantified in the input cohort and GFP-positive cohort. The ratio of GFP-positive to input cells for a given variant is normalized against BRCA1 synonymous variants in the BRCA1 siRNA condition. Image created with BioRender.com.

(FRT) for integration and expression of one BRCA1 variant per cell [11]. We thus create three libraries of cells, each cell carrying a single variant integrated at the FRT site.

Cells containing the library of BRCA1 variants integrated into the FRT site were subjected to two transfections, two days apart, of either control siRNA or siRNA targeting the 3'UTR of endogenous BRCA1 mRNA (Fig 1B). Concurrent with the second transfection, a plasmid expressing the I-Sce1 enzyme was transfected to create a double-stranded break within a gene encoding GFP. In the control siRNA condition, BRCA1 WT is present and can repair the double-stranded break via homology directed repair (HDR) and correct the defect in GFP, converting the cell to GFP-positive. In the BRCA1 siRNA condition, if the integrated variant can repair the double-stranded break through HDR, then the cell will convert to GFP-positive. Conversely, if the integrated variant cannot repair the double-stranded break via HDR, then the cell will remain GFP-negative. Cells were checked for GFP-positive percentage two days after the second transfection, and if the mutagenic library sample had a decreased GFP-positive percentage compared to that of the control siRNA sample (S1 Fig), then the experiment proceeded with cell sorting the following day. As there are loss of function variants expected in each mutagenic library, the library sample should have a lower GFP percentage than that of the control siRNA, in which the BRCA1 WT is expressed and can repair the double-stranded break regardless of the function of the integrated variant. qRT-PCR (S2 Fig) was performed to confirm the knockdown of endogenous BRCA1 through siBRCA1 3'UTR, and western blots (S3 Fig) were performed to confirm knockdown of BRCA1 with siBRCA1 CDS.

## Determination of BRCA1 variant function in DNA repair from assay score

All three libraries were combined for the purpose of establishing score cutoffs for categorizing loss of function (LOF), intermediate function, or functionally normal for a given variant. To establish score cutoffs, the pool was divided into three categories of missense, nonsense, and synonymous variants (Fig 2, S3 and S10 Tables in S1 Data). In the control siRNA condition, with endogenous wild-type BRCA1 present in all cells, all three variant cohorts had scores that centered around a functional score of zero indicating functionally normal. In the BRCA1 3'UTR siRNA experimental condition, endogenous BRCA1 expression is silenced, and the cell

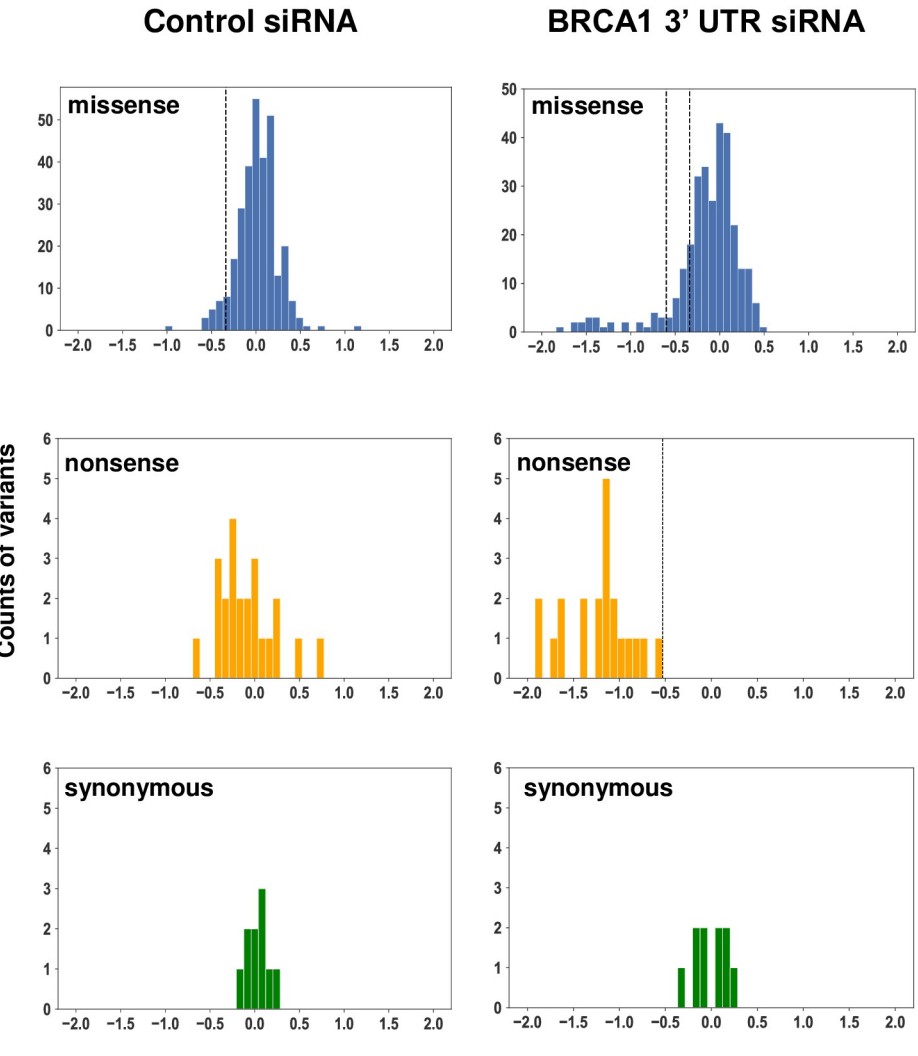

**Fig 2. Distribution of multiplexed functional scores for BRCA1 variants.** Three cohorts of variants—missense, nonsense, and synonymous—are plotted by functional score on a histogram in control siRNA (left) and BRCA1 3'UTR siRNA conditions. The cutoff between functionally normal and intermediate function is established by modeling the missense variants in the control siRNA condition as a normal distribution and picking the value of the first percentile of variants. The cutoff between nonfunctional and intermediate function was established by picking the hundredths place just above the highest scoring nonsense variant in the BRCA1 3'UTR siRNA condition. Variants scoring between these two cutoffs are classified as intermediate function.

is dependent on the integrated variant BRCA1 to repair the double-stranded breaks. In this condition, the scores for nonsense variants shifted to the left, scores for synonymous variants remained centered around zero, and scores for missense variants separated into one group that remained centered around zero (functionally normal) and a second, smaller group with scores that shifted to the left (loss of function). To classify missense variants for function in the HDR assay, we set thresholds in the functional scores. The score cutoff delineating functionally normal from intermediate function (-0.338) was determined through the modeling of the missense variants in the control siRNA condition as a normal distribution and selecting the value representing the bottom first percentile of missense variants. The score cutoff delineating non-function from intermediate function (-0.60) was selected as the hundredths place above the highest scoring nonsense variant in the BRCA1 siRNA condition. Additionally, variants with a replicate variance score greater than one were excluded from inclusion in the final dataset (S4 Fig).

Nucleotide changes and assay scores for synonymous variants are available in S10 Table in S1 Data, and amino acid changes and assay scores for missense and nonsense variants are available in S3 Table in S1 Data.

## Singleton assay for selected variants

We obtained a lower number of variants for this region of BRCA1 than was anticipated by prior studies [11–13]. One issue was the density of the data on which functional interpretations were made. The number of variants included in the analysis that were functionally interpreted was determined by the read count cutoff. The sparsity of the dataset created uncertainty regarding optimal read count cutoff to delineate signal from "noise" in our multiplexed results. The read count cutoff was initially set low to include a high number of variants to address the sparse nature of the dataset. In particular, there were more LOF variants scored than were anticipated, and this would be expected if the read-count threshold for including datapoints was too low and included noise along with results. We tested in the singleton HDR assay many of the LOF variants determined using the multiplexed HDR assay to see if the results were consistent.

We employed the singleton assay to delineate signal from noise in our multiplexed results, plus we tested some codons of interest that were missed in the multiplex assay. These latter codons/amino acid residues were on the PALB2 binding surface as well as select serines that may be targeted for phosphorylation. A total of 25 variants, 19 of which were also tested in the multiplexed assay, were evaluated in the singleton HDR assay (Fig 3, S4 Table in S1 Data). In the singleton assay, there is no BRCA1 variant integrated into the cell, but the variant being tested is expressed by transient transfection of the appropriate plasmid. Each variant was tested in triplicate alongside (1) a control siRNA condition with control plasmid transiently expressed, (2) the BRCA1 3'UTR siRNA targeting endogenous BRCA1 along with control plasmid, and (3) the BRCA1 3'UTR siRNA and a transiently expressed plasmid encoding BRCA1 WT. Of the 25 variants tested, two were nonfunctional, two were intermediate function, and 21 were functional. Of the 19 variants also tested in the multiplexed assay, one variant, BRCA1 p.Met1400Ser, scored as loss of function in both the singleton and multiplexed assays. Two variants, BRCA1 p.Leu1455Ile and BRCA1 p.Ile1465Leu, scored as intermediate function in the singleton assay and loss of function in the multiplex assay. A total of 16 variants scored as functionally normal in the singleton assay, and loss of function in the multiplexed assay. This discordance between singleton and multiplexed results indicated that the initial read count cutoffs for the multiplexed assay were too low, indicating that an unacceptably high amount of noise was present in the results. To address this, the read count was raised to

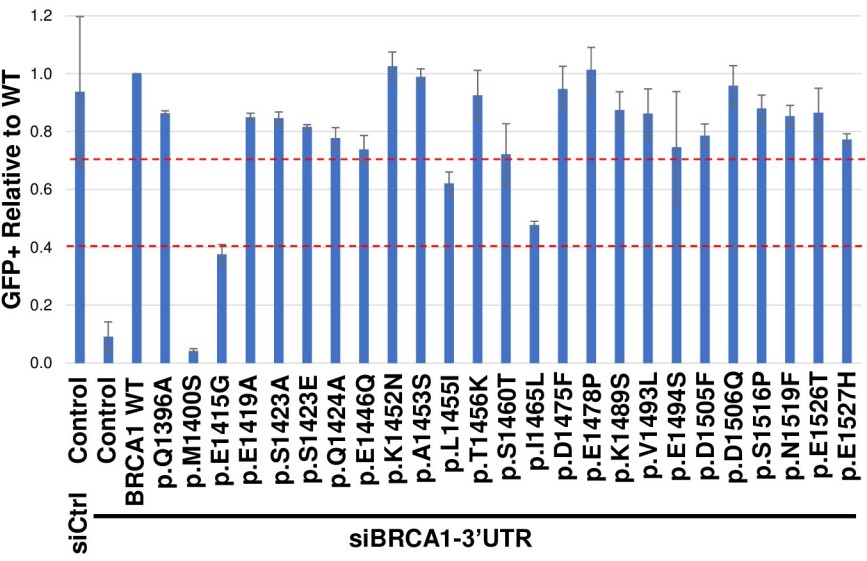

**Fig 3. Selected BRCA1 variants tested in the singleton HDR assay.** A total of 25 variants were tested in the singleton assay, which tests the same function as the multiplexed assay but with one variant at a time instead of hundreds of variants simultaneously. For the singleton assay, the cutoff between nonfunctional and intermediate function is 0.4 (-1.32 once $\log_2$ transformed), and the cutoff between intermediate function and functionally normal is 0.7 (-0.51 once $\log_2$ transformed); these values were indicated with the red dashed lines. Of the 25 variants tested, two were LOF, two were intermediate function, and 21 were functionally normal.

eliminate 15 of the 18 discordant results. Of the six variants tested in the singleton assay but not the multiplexed assay, one variant (BRCA1 p.Glu1415Gly) scored as loss of function, and five variants (BRCA1 p.Gln1396Ala, BRCA1 p.Glu1415Ala, BRCA1 p.Ser1423Ala, BRCA1 p. Ser1423Glu, and BRCA1 p.Gln1424Ala) scored as functionally normal. Multiplexed results presented in Fig 2 and following were using the higher read-count threshold determined in this analysis. Applying higher read count thresholds resulted in a total of 310 variants evaluated in the multiplexed assay, five of which were also tested in the singleton. A sixth variant included in the final multiplexed dataset, BRCA1 p.Val1378Ile, was tested in the singleton assay previously by our lab [35] and that singleton result was included in this analysis. This left a cohort of 20 variants tested in the singleton assay but not included in the final multiplexed results, for a total of 330 variants evaluated. Compiled multiplexed and singleton results are available in S2 Table in S1 Data.

## Comparison of multiplexed results to clinical classification

Of the five variants evaluated in the multiplexed assay that were designated by ClinVar as benign or likely benign, all five scored as functionally normal (Fig 4). Of the 11 variants evaluated in the multiplexed assay that were designated by ClinVar as pathogenic or likely pathogenic, all 11 scored as loss of function. It is worth noting that all 11 pathogenic or likely pathogenic variants were nonsense variants. Since all five of the benign variants captured in this dataset were functional, the specificity of this assay was 100% for a functionally normal variant to be benign or likely benign. This is comparable to the specificity of two previous sets of data published for the multiplex HDR assay using BRCA1–81% specificity in the same multiplexed HDR assay and 100% specificity in a multiplexed cisplatin resistance assay for evaluation of BRCT domain missense and nonsense variants [13], and 100% specificity for the same multiplexed evaluation of HDR function of RING domain and N-terminal variants

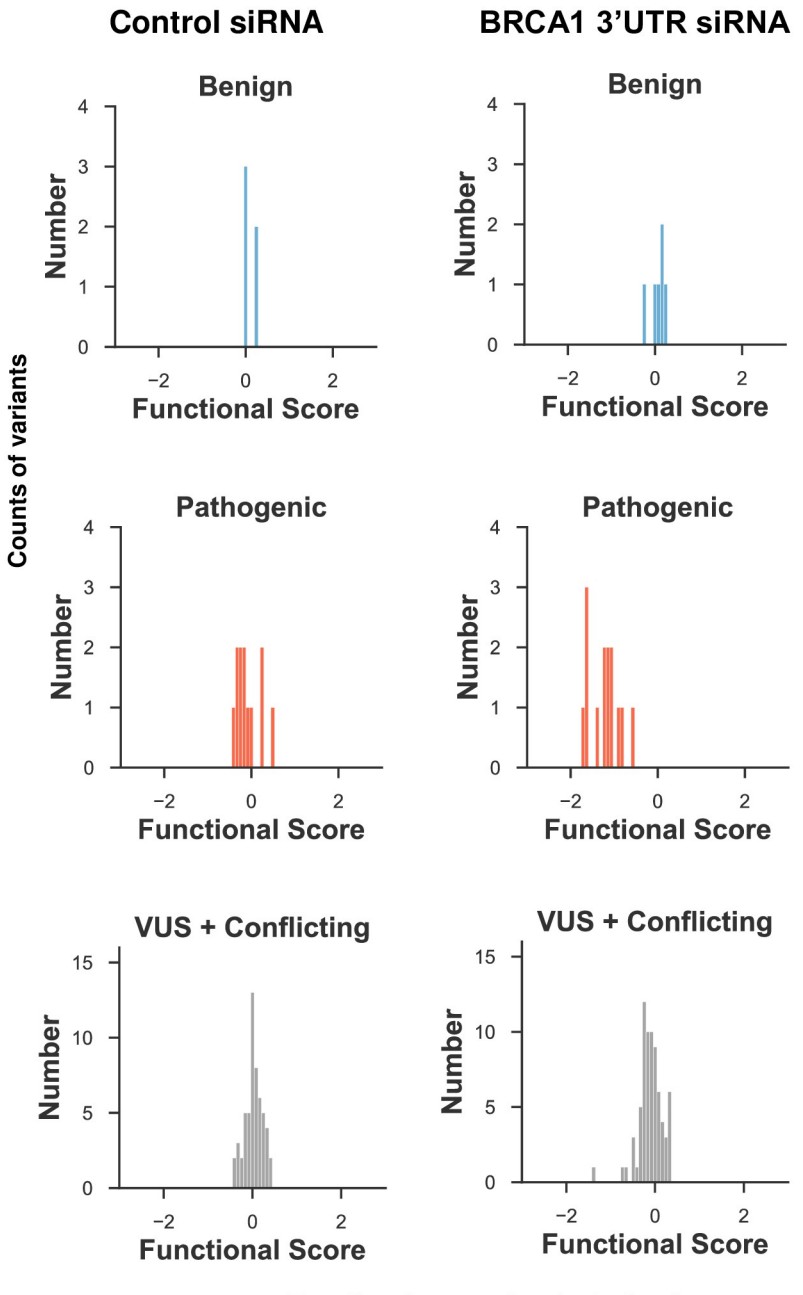

**Fig 4. Comparison of functional scores from multiplexed HDR assay to variant classification in the ClinVar database.** Variants with ClinVar designations of benign or likely benign (top) pathogenic or likely pathogenic (middle) and variants of uncertain significance (VUS) or conflicting interpretations (bottom) are plotted on histograms by their DNA repair functional score. Results are from assays under control conditions with endogenous BRCA1 protein present (left) or from siBRCA1 3'-UTR conditions (right).

[12]. Similarly, since the LOF variants identified in the HDR assay were consistent with the clinical classification of pathogenic or likely pathogenic, the sensitivity of this assay was 100% for a LOF variant to be pathogenic or likely pathogenic. This is comparable to the sensitivity of two previous sets of data published for the multiplex HDR assay—93% sensitivity

in the multiplexed HDR assay and 86% sensitivity of the multiplexed cisplatin resistance assay for evaluation of BRCT domain missense and nonsense variants [13] and 93% sensitivity for the same multiplexed evaluation of HDR function of RING domain and amino-terminal variants [12]. Of the 66 variants evaluated in the multiplexed assay that were designated in ClinVar as VUS or conflicting, 59 scored as functionally normal, five scored as intermediate function, and two scored as loss of function. This percentage (3.0%, 2 of 66) of VUS and conflicting that scored as loss of function was lower than the corresponding percentage (5.8%, 14 of 240) of the RING domain and amino-terminal evaluation [12]. OddsPath analysis [19] yielded classifications for interpreting the functional data as BS3_moderate and PS3_moderate. The conflicting variants evaluated in this manuscript contained entries asserting a classification of Uncertain Significance, Likely Benign, or Benign. Details regarding ClinVar classifications, including year of last evaluation, number of stars in ClinVar entry, and number of classifications for conflicting variants are found in S11 Table in S1 Data.

While the variants tested are the same variants in ClinVar at the amino acid level, at the nucleotide-level variants may differ between the 330 variants and the corresponding ClinVar entry. Some ClinVar variants have multiple entries that differ at the nucleotide level but not at the amino acid level. Some of the 330 variants may have multiple barcodes corresponding to different nucleotide changes that result in the same amino acid change. Details regarding the multiplexed variant nucleotide-level changes can be found in the long-read sequencing legend pairing barcodes to variants on the Gene Expression Omnibus dataset. We did not include the nucleotide changes for the missense and nonsense variants in the supplemental tables since the variants used in this study were not expressed from the *BRCA1* genomic locus but rather from a plasmid encoding the BRCA1 cDNA integrated at a Flp-In recombination site.

## Sequence-function map of multiplexed results

After final read count cutoffs were established, a total of 310 variants were evaluated in the multiplex assay (S3 Table in S1 Data), as well as six variants that were evaluated in the singleton but not the multiplex assay (S6 Fig). Of the 310 multiplexed variants, 236 scored as functionally normal, 27 scored as intermediate function, and 47 scored as loss of function. Of the 47 loss of function variants, 21 were nonsense and 26 were missense. All nonsense variants scored as loss of function, which is expected and is an indicator of the accuracy of the assay.

As BRCA1-PALB2 interaction is important in homology-directed repair, it was expected that variants that were among the BRCA1 residues known to interact with PALB2 would have an increased likelihood of scoring as loss of function. In BRCA1 residues 1364–1437, which interact with PALB2 [34, 36], there were 35 missense variants evaluated. Of the 35 missense variants, 12 (34%) scored as loss of function, compared with 14 of 254 (6%) missense variants outside of this region. Of the 14 nonfunctional missense variants outside of the PALB2 binding region, five (BRCA1-p.Thr1456Lys, BRCA1-p.Ser1460Thr, BRCA1-p.Ser1460Arg, BRCA1-p.Ser1483Arg, and BRCA1-p.Tyr1509Ser) are located at WT residues that can be phosphorylated (serine, threonine, or tyrosine), and nine are outside of those residues and may indicate protein interactions that are yet unknown. Of the five possible phosphorylation sites corresponding to four residues, none are part of an SQ/TQ motif. While BRCA1-p.Ser1460 [37–39] and BRCA1-p.Ser1483 [40] phosphorylation has been documented, there is no clear conclusion as to the functional impact of the phosphorylation.

## Discussion

### Summary

In this study, we applied the multiplex HDR analysis for DNA repair function to BRCA1 residues 1280–1576. This is the domain of BRCA1 that is phosphorylated at multiple sites following DNA damage [26–31] and binds to PALB2, the protein that physically bridges BRCA1 with BRCA2 and RAD51 [34, 41–43]. With this dataset, 310 variants evaluated in the multiplex assay and 20 previously unevaluated variants evaluated in the singleton assay gave a total of 330 missense and nonsense variants evaluated for function in HDR. A multiplexed evaluation of residues surrounding and including the BRCA1 coiled-coil region has not previously been conducted. Nonfunctional variants were present in higher numbers in the residues known to interact with the PALB2 protein. Our results of LOF variants in this 74-residue region highlight the functional importance of BRCA1-PALB2 binding.

Variants in the 297-residue region tested in this study have been shown to have reduced HDR activity, reduced PALB2 interaction, reduced protein stability, and increased cisplatin and olaparib sensitivity [36, 44–46]. The identification of BRCA1 variants that affect DNA repair function is of particular clinical importance as synthetic lethality between BRCA1 LOF genes and sensitivity to DNA damaging agents, such as PARP inhibitors, cisplatin, or topoisomerase inhibitors is well documented [7–9]. Additionally, the murine homologue of p.BRCA1-- Leu1407Pro has been shown to promote tumor formation in mice [32]. Indeed, a different substitution at this residue, BRCA1-p.Leu1407Ala, scored as LOF in our multiplexed assay.

### Clinical relevance of variants in the coiled-coil domain

Classifying VUS as benign or pathogenic is critical for improving cancer surveillance and treatment options for patients. Classifying VUS as benign may prevent an unnecessary surgery from occurring, while classifying VUS as pathogenic may allow for early detection of cancer or increase available treatment options, such as recommending a PARP inhibitor or platinum drug for patients with a nonfunctional HDR-related protein. The ClinVar database lists pathogenic or likely pathogenic missense variants in this 297-codon region at residue 1495 (BRCA1-p.Arg1495Lys, BRCA1-p.Arg1495Met, and BRCA1-p.Arg1495Thr) and residue 1559 (BRCA1-p.Glu1559Gln and BRCA1-p.Glu1559Lys). BRCA1-p.Arg1495Met is known to affect splicing [47], and due to the similar location at exon-intron boundaries, it is implied that the remainder of the variants at residue 1495, as well as the variants at residue 1559, also result in differential splicing, leading to its ClinVar classification of pathogenic or likely pathogenic. BRCA1-p.Glu1559Val was evaluated and scored as intermediate function, but the multiplex HDR assay uses pre-spliced cDNA to encode BRCA1 variants and is independent of splicing effects. Also, two missense variants that presumably do not affect splicing, BRCA1-p. Gln1408His and BRCA1-p.Met1411Thr, are listed in ClinVar as pathogenic or likely pathogenic. As both of these variants occur within the region known to interact with PALB2, the pathogenic nature of the variant may be due to decreased BRCA1-PALB2 binding. This again emphasizes the importance of the BRCA1-PALB2 interface.

While the RING and BRCT domains have dozens of pathogenic or likely pathogenic missense variants documented in ClinVar, the coiled-coil domain only documents the two non-splicing variants indicated above. One potential explanation for the lack of more pathogenic missense variants in the coiled-coil domain is the possibility of BRCA1-independent HDR mediated by PALB2 [48]. It is possible that the HDR process in the HeLa cell line in use in this study is dependent on the PALB2-BRCA1 interaction, but in other cells HDR may be independent of the PALB2 binding to BRCA1.

## Dataset challenges and opportunities

The biggest obstacle facing this study was the sparseness of the dataset. Overall, 5.6% of possible missense and nonsense variants for this region were tested in this assay. The main two causes of sparsity were the low variant creation percentage in two of the plasmid libraries and low integration percentage of the plasmid library into the cell line. The low integration percentage highlights the inefficiency of the FRT / Flp-In recombination system and the difficulty in applying this recombination site for integrating variant libraries. While this integration system has been used in the multiplexed analysis of BRCA1 [11–13], the FRT is best used for recombination of single variants. The more efficient Bxb1 recombination system [49] is perhaps better for integrating variant libraries into cells.

Combined with two previous publications from our lab [12, 13], we have now probed residues 1–302 and 1280–1863 of BRCA1 using a multiplexed HDR assay, as well as residues 1577–1863 using a multiplexed cisplatin resistance assay. Concordance among singleton assay and multiplexed assay results, as well as concordance with other assays that test other functions of the BRCA1 protein, should reliably indicate the functional impact of missense and nonsense variants on BRCA1. A BRCA1 variant that results in loss of function in DNA repair should indicate higher likelihood of cancer predisposition.

## Supporting information

**S1 Fig. Functional check of GFP percentage on Day 5 of multiplexed assay.** Before sorting into GFP-positive and negative cohorts and extracting genomic DNA on Day 6, different siRNA conditions were evaluated for GFP-positive percentage on Day 5 of the multiplexed assay. siBRCA1 3'UTR samples scored a lower percentage than siControl, as it is expected that each library will have some loss of function variants. siBRCA1 coding specific (CDS) scored low as both the endogenous BRCA1 and integrated BRCA1 variant were silenced.
(TIF)

**S2 Fig. qPCR values for multiplexed replicates.** Examples are shown for endogenous BRCA1 expression for siControl and siBRCA1 3'UTR treated samples in each multiplexed replicate.
(TIF)

**S3 Fig. Western blots for multiplexed replicates.** Examples are shown for the analysis of BRCA1 expression for siControl, siBRCA1 3'UTR, and siBRCA1 CDS samples for one replicate per library. RHA1 is an antibody specific for RNA Helicase A and detects a 140 kDa protein used as a loading control.
(TIF)

**S4 Fig. Functional score variance of variants between replicates.** Variants are plotted by their functional score in the HDR assay against variance of the score between replicates. Any variants with a variance greater than 1 were not included in the final dataset.
(TIF)

**S5 Fig. Comparison of variants with ClinVar designation to multiplexed categorization.** Of the five variants with a benign or likely benign ClinVar designation, all five scored as functionally normal in the multiplexed assay, indicating 100% specificity. Of the 11 variants with a pathogenic or likely pathogenic ClinVar designation, all 11 scored as LOF in the HDR assay, indicating 100% sensitivity.
(TIF)

**S6 Fig. Sequence-Function map of 310 missense and nonsense variants.** After categorization as LOF, intermediate function, or functionally normal, variants were graphed according to

their position on the BRCA1 protein (x-axis) and the substituted amino acid (y-axis). Red indicates LOF, pink indicates intermediate function, and white indicates functionally normal. Black outlines with a black dot indicate WT amino acid, and gray indicates insufficient data to score the function.
(TIF)

**S1 Raw images. Full western blots.** Immunoblots used in S3 Fig are shown for the full membrane. BRCA1 and RHA1 stains for CC1 replicate 3, CC2 replicate 2, CC3 replicate 3, and CC3 replicate 4.
(PDF)

**S1 Data. BRCA1CC updated 2023 October 16.xlsx.** The following tables are provided as supplementary information: S1 Table. Table of Contents, S2 Table. Compiled Scores of 330 Variants, S3 Table. Multiplexed Scores of 310 Variants, S4 Table. Singleton Scores of 25 Variants, S5 Table. Full List of mutagenic primers used in multiplexed assay—297 fwd and 297 rev primers, S6 Table. Primers for singleton variant construction, other plasmid construction, qPCR, sequencing, and nested PCR, S7 Table. Functional results (GFP+%) on Day 5 of multiplexed assay, S8 Table. Endogenous BRCA1 qPCR results, S9 Table. Counts of sorted cell cohorts in multiplexed assay, S10 Table. Synonymous Variant Nucleotide Changes and Counts, S11 Table. Details of variant ClinVar listings.
(XLSX)

## Acknowledgments

We thank Dr. Amanda Toland for thoughtful edits to the manuscript. Thank you to the genomics and flow cytometry cores (P30CA016058) at the Ohio State University, the PacBio sequencing services at University of Washington, and the Steve and Cindy Rasmussen Institute for Genomic Medicine genomic services core at Nationwide Children's Hospital (Nationwide Foundation Pediatric Innovation Fund).

## Author Contributions

**Conceptualization:** Gregory Nagy, Aleksandra I. Adamovich, Lea M. Starita, Jeffrey D. Parvin.

**Data curation:** Mariame Diabate, Nahum Smith, Hyeongseon Jeon.

**Formal analysis:** Mariame Diabate, Nahum Smith, Hyeongseon Jeon.

**Funding acquisition:** Lea M. Starita, Jeffrey D. Parvin.

**Investigation:** Gregory Nagy, Tapahsama Banerjee, Aleksandra I. Adamovich, Shruti Dhar, Wenfang Liu, Katherine Burgess.

**Methodology:** Lea M. Starita, Jeffrey D. Parvin.

**Project administration:** Lea M. Starita, Jeffrey D. Parvin.

**Resources:** Lea M. Starita, Jeffrey D. Parvin.

**Supervision:** Gregory Nagy, Tapahsama Banerjee, Aleksandra I. Adamovich, Dongjun Chung, Jeffrey D. Parvin.

**Validation:** Nahum Smith, Hyeongseon Jeon.

**Visualization:** Gregory Nagy, Mariame Diabate.

**Writing – original draft:** Gregory Nagy.

**Writing – review & editing:** Gregory Nagy, Aleksandra I. Adamovich, Jeffrey D. Parvin.

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
