## [Decision Letter · Decision Letter 0]

14 Aug 2023

PONE-D-23-19626Multiplexed assay of variant effect reveals residues of functional importance in the BRCA1 coiled-coil and serine cluster domainsPLOS ONE

Dear Dr. Parvin,

Thank you for submitting your manuscript to PLOS ONE. After careful consideration, we feel that it has merit but does not fully meet PLOS ONE’s publication criteria as it currently stands. Therefore, we invite you to submit a revised version of the manuscript that addresses the points raised during the review process.

We look forward to receiving your revised manuscript.

Kind regards,

Alvaro Galli

Academic Editor

PLOS ONE

2. In your Methods section, please report the source of HeLa cells used for your study.

“We thank Dr. Amanda Toland for thoughtful edits to the manuscript. Thank you to the genomics and flow cytometry cores (**P30CA016058)** at the Ohio State University, the PacBio sequencing services at University of Washington, and the Steve and Cindy Rasmussen Institute for Genomic Medicine genomic services core at Nationwide Children’s Hospital (Nationwide Foundation Pediatric Innovation Fund). This work was funded in part by NIH R01 CA228083 to JDP and LMS, NIH R01 CA228083–01A1S1 to MD, and a University Fellowship, NIH T32GM068412-11A1, and a Pelotonia Fellowship to GN.”

“JDP and LMS were supported by R01 CA228083 from the National Cancer Institute.

MD was supported by R01 supplement CA228083-01A1S1 from the National Cancer Institute

GN was supported by T32 GM068412 from the National Institute of General Medical Sciences and a Pelotonia Fellowship.

Reviewers' comments:

Reviewer's Responses to Questions

**Comments to the Author**

1. Is the manuscript technically sound, and do the data support the conclusions?

Reviewer #1: Yes

2. Has the statistical analysis been performed appropriately and rigorously? 

Reviewer #1: Yes

3. Have the authors made all data underlying the findings in their manuscript fully available?

Reviewer #1: Yes

4. Is the manuscript presented in an intelligible fashion and written in standard English?

Reviewer #1: Yes

5. Review Comments to the Author

Reviewer #1: Ascertaining which of the many BRCA1 “variants-of-uncertain-significance” (VUSs) does or does not impair BRCA1 tumor suppression activity is an important clinical issue. Although BRCA1 has been implicated in a number of cellular processes, its ability to promote homology-directed repair (HDR) appears to correlate quite well with its tumor suppression activity. Thus, in recent years, the authors and other groups have designed multiplexed strategies that allow efficient examination of the HDR activities of many BRCA1 variants, including those already observed in patients (e.g., in the ClinVar database) and those that might arise in the future. Initially, these studies focused on variants within the well-defined N-terminal RING and C-terminal BRCT domains of the BRCA1 protein. In this study, Nagy et al. examine variants within an internal region of BRCA1 (residues 1280-1576), which includes the binding site for the PALB2 protein (residues 1364-1437) as well an intriguing, but poorly defined, cluster of “SQ/TQ” phosphorylation sites for the damage-inducible ATR/ATM kinases.

The manuscript readily fulfills all the criteria necessary for publication in PLoS ONE. As one might expect given their previous work in the development and use of multiplexed HDR assays, the experimental and statistical analyses are highly rigorous and consequently the data generated are very convincing. Only a fraction of the possible variants within this region (BRCA1 residues 1280-1576) are examined in this study: at least from my calculations (which may not be entirely correct), 81 of the 350 ClinVar variants and 310 of the 5940 total possible variants were evaluated. Nonetheless, the data generated from this initial survey have potential clinical significance and might also yield clues about the molecular mechanisms of BRCA1 function. Having said that, the manuscript would be improved by presenting some of the data in a more accessible manner and by discussing the significance of certain variants.

Comments:

1. The authors should provide a simple table listing 1) each variant tested (310?), 2) whether it was tested by the multiplexed and/or singlet assay, 3) the observed HDR activity (i.e., function, intermediate, nonfunctional), and 4) whether it is currently listed in the ClinVar database. It may be possible to ferret out all this information by working through the current tables, but it would be a rather painful exercise.

2. This segment of BRCA1 includes the Palb2-binding sequences, and the authors’ data nicely underscore the important of this interaction in HDR and tumor suppression. However, this segment also encompasses the poorly-understood, but intriguing, cluster of SQ/TQ phosphorylation sites recognized by the ATR/ATM kinases. Interestingly, on lines 319-321, the authors mention that five of the nonfunctional missense variants affect either serine, threonine, or tyrosine. However, they don’t identify them. Given their potential significance, the specific variants should be identified in the text and noted as to whether they correspond to SQ/TQ sites. And if so, then some discussion of the literature would be warranted, such as whether the site is known to be phosphorylated by ATR/ATM and whether phosphorylation has functional consequences.

3. Is Figure 5 necessary? At least for me, it doesn’t provide any information.

4. Do the authors wish to mention “Figure S4” in the sentence on lines 223-226?

5. On lines 347-350, the authors state that “The ClinVar database lists pathogenic or 348 likely pathogenic missense variants in this 297-codon region that affect splicing at residue 1495 and residue 1559”. Is there a reference describing these splicing results?

6. PLOS authors have the option to publish the peer review history of their article (what does this mean?). If published, this will include your full peer review and any attached files.

Reviewer #1: No

---

## [Author Response · Author response to Decision Letter 0]

7 Sep 2023

Please see Response to Reviewer document

---

## [Decision Letter · Decision Letter 1]

20 Sep 2023

PONE-D-23-19626R1Multiplexed assay of variant effect reveals residues of functional importance in the BRCA1 coiled-coil and serine cluster domainsPLOS ONE

Dear Dr. Parvin,

Thank you for submitting your manuscript to PLOS ONE. After careful consideration, we feel that it has merit but does not fully meet PLOS ONE’s publication criteria as it currently stands. Therefore, we invite you to submit a revised version of the manuscript that addresses the points raised during the review process.

We look forward to receiving your revised manuscript.

Kind regards,

Alvaro Galli

Academic Editor

PLOS ONE

Journal Requirements:

Reviewers' comments:

Reviewer's Responses to Questions

**Comments to the Author**

1. If the authors have adequately addressed your comments raised in a previous round of review and you feel that this manuscript is now acceptable for publication, you may indicate that here to bypass the “Comments to the Author” section, enter your conflict of interest statement in the “Confidential to Editor” section, and submit your "Accept" recommendation.

Reviewer #1: All comments have been addressed

Reviewer #2: (No Response)

2. Is the manuscript technically sound, and do the data support the conclusions?

Reviewer #1: Yes

Reviewer #2: Yes

3. Has the statistical analysis been performed appropriately and rigorously? 

Reviewer #1: Yes

Reviewer #2: Yes

4. Have the authors made all data underlying the findings in their manuscript fully available?

Reviewer #1: Yes

Reviewer #2: Yes

5. Is the manuscript presented in an intelligible fashion and written in standard English?

Reviewer #1: Yes

Reviewer #2: Yes

6. Review Comments to the Author

Reviewer #1: (No Response)

Reviewer #2: In this manuscript submitted as a research article to Plos One, the authors have performed a functional multiplexed assay measuring the DNA repair activity of BRCA1 variants located in the coiled-coil and serine cluster domain, i.e. encompassing amino acid residues 1280 and 1576. The results were compared with functional singleton assays (studying one variant at a time) and also to available ClinVar database variant interpretations of the same variants. Functional assays, as the homology-directed repair (HDR) reporter assay used in this study, is a validated assay supported by the BRCA1 Variant Curation Expert Panel guidelines, like CanVig-UK or Enigma, and variants that are either loss of function or functional in the HDR assay may be scored in pathogenic or benign direction by using either the PS3 or BS3 ACMG-AMP subscores, respectively. Even though BRCA1 is a well-characterized gene, interpretation of BRCA1 variants is still challenging for the clinical laboratories. As discussed by the authors, when a variant of unknown significance (VUS) is detected, no clinical management decisions can therefore be made based on the finding. This makes the surveillance and management of VUS carriers particularly challenging, and the classification of a VUS is a significant barrier for the clinicians. Therefore, this manuscript will be especially useful and valuable for clinical/laboratory geneticist working with variant interpretation of rare BRCA1 variants detected in patients with suspected hereditary breast and ovarian cancer. Anyhow, I have some minor points that need to be addressed before this manuscript can be published by Plos One.

Minor revision:

1. The authors do not mention how functional analyses of missense variants are used in the variant interpretation guidelines following the original ACMG-AMP guidelines (Richards, 2015. PMID: 25741868), or by the more newly released BRCA1 specific guidelines published by CanVIG-UK (https://www.cangene-canvaruk.org/_files/ugd/ed948a_f05b6f7767ad499a8539131d06035432.pdf) or ClinGene/Enigma (https://cspec.genome.network/cspec/ui/svi/doc/GN092). This topic should be introduced in the introduction.

2. Line 74: In this 297-residue region there are greater than 350 missense VUS in ClinVar. Change to: In this 297-residue region there are more than 350 missense VUS in ClinVar.

3. Line 222-226: Cells were sorted based on GFP fluorescence, and genomic DNA was subsequently extracted to identify the integrated BRCA1 variant in each cell. The frequencies of variants in the GFP-positive and input cohorts were determined and a functional score for each variant relative to WT was calculated based on variant distribution between the GFP-positive and input cohorts using Enrich2 software (20). This paragraph belongs to the Method section in my opinion.

4. Line 231: The synonymous variants they used to define “normal” thresholds, are they listed somewhere? If not, please do so. Did the authors check that these variants do not affect splicing or do not involve conserved nucleotides?

5. The authors used ClinVar classifications to compare the functional multiplexed results, but did the authors include all Clin Var classifications regardless of i) year of classification, ii) amount of ClinVar stars, iii) where any ClinVar classifications by expert panels? It would be nice if this information could be included in Suppl Table 2 and also in the Methods section. The same question goes for Figure 4 (conflicting interpretations).

6. Line 310: This percentage (3.0%, 2 of 66) of VUS and conflicting that scored as loss of function was lower than the corresponding percentage (5.8%, 14 of 240) of the RING domain and amino-terminal evaluation (12). The authors need to specify what they mean by conflicting? Conflicting between which categories, VUS vs benign, VUS vs likely benign, VUS vs likely pathogenic, etc?

7. Figure 3: Could they mark their cutoff thresholds (0.4 and 0.7) as lines in the figure?

8. Suppl table S2: This is a very useful table and will be used by clinical/laboratory geneticist working with variant interpretation of rare BRCA1 variants. Could the authors include which reference transcript for BRCA1 they are referring to and also include the nomenclature at the DNA level?

For instance: Q1281* should be written as BRCA1 (NM_007294.4): c.3841C>T Q1281* or c.3841C>T Gln1281* (Three –letter amino acid codes are preferred by HGVS guidelines (Sequence Variant Nomenclature (hgvs.org)).

7. PLOS authors have the option to publish the peer review history of their article (what does this mean?). If published, this will include your full peer review and any attached files.

Reviewer #1: No

Reviewer #2: No

---

## [Author Response · Author response to Decision Letter 1]

3 Oct 2023

Please see word file with itemized response to reviewers. We thank the reviewers for their consideration of this study and appreciate recommended changes that have improved the presentation of the manuscript.

---

## [Decision Letter · Decision Letter 2]

12 Oct 2023

Multiplexed assay of variant effect reveals residues of functional importance in the BRCA1 coiled-coil and serine cluster domains

PONE-D-23-19626R2

Dear Dr. Parvin,

We’re pleased to inform you that your manuscript has been judged scientifically suitable for publication and will be formally accepted for publication once it meets all outstanding technical requirements.

Kind regards,

Alvaro Galli

Academic Editor

PLOS ONE

Additional Editor Comments (optional):

Reviewers' comments:

Reviewer's Responses to Questions

**Comments to the Author**

1. If the authors have adequately addressed your comments raised in a previous round of review and you feel that this manuscript is now acceptable for publication, you may indicate that here to bypass the “Comments to the Author” section, enter your conflict of interest statement in the “Confidential to Editor” section, and submit your "Accept" recommendation.

Reviewer #1: All comments have been addressed

Reviewer #2: All comments have been addressed

2. Is the manuscript technically sound, and do the data support the conclusions?

Reviewer #1: Yes

Reviewer #2: Yes

3. Has the statistical analysis been performed appropriately and rigorously? 

Reviewer #1: Yes

Reviewer #2: Yes

4. Have the authors made all data underlying the findings in their manuscript fully available?

Reviewer #1: Yes

Reviewer #2: Yes

5. Is the manuscript presented in an intelligible fashion and written in standard English?

Reviewer #1: Yes

Reviewer #2: Yes

6. Review Comments to the Author

Reviewer #1: (No Response)

Reviewer #2: The authors have adequately addressed my comments raised in the previous round of review and are now ready for publication. Just a small comment, in Table S10 the correct nomenclature for synonymous variants according to HGVS is c.5007C>A p.(Ala1669=), rather than c.5007C>A p.(=). Please make these changes in the table before publication.

7. PLOS authors have the option to publish the peer review history of their article (what does this mean?). If published, this will include your full peer review and any attached files.

Reviewer #1: No

Reviewer #2: No

---

## [Editor Report · Acceptance letter]

24 Oct 2023

PONE-D-23-19626R2 

Multiplexed assay of variant effect reveals residues of functional importance in the *BRCA1* coiled-coil and serine cluster domains 

Dear Dr. Parvin:

I'm pleased to inform you that your manuscript has been deemed suitable for publication in PLOS ONE. Congratulations! Your manuscript is now with our production department. 

Kind regards, 

on behalf of

Dr. Alvaro Galli 

Academic Editor

PLOS ONE